# Auction-Based Consensus of Autonomous Vehicles for Multi-Target Dynamic Task Allocation and Path Planning in an Unknown Obstacle Environment

**Wan-Yu Yu** [1], **Xiao-Qiang Huang** [2], **Hung-Yi Luo** [1], **Von-Wun Soo** [2] and **Yung-Lung Lee** [3,*]

1   Institute of Information Systems and Applications, National Tsing Hua University, 101, Section 2, Kuang-Fu Road, Hsinchu 30043, Taiwan; s9865804@m98.nthu.edu.tw (W.-Y.Y.); sai28055707@gmail.com (H.-Y.L.)
2   Department of Computer Science, National Tsing Hua University, 101, Section 2, Kuang-Fu Road, Hsinchu 30043, Taiwan; xqhuang718@gmail.com (X.-Q.H.); soo@cs.nthu.edu.tw (V-W.S.)
3   Department of Power Vehicle and Systems Engineering, Chung Cheng Institute of Technology, National Defense University, No. 75, Shiyuan Rd., Dashi Jen 335, Taoyuan, Taiwan
*   Correspondence: yunglunglee84@ccit.ndu.edu.tw

**Abstract:** The autonomous vehicle technology has recently been developed rapidly in a wide variety of applications. However, coordinating a team of autonomous vehicles to complete missions in an unknown and changing environment has been a challenging and complicated task. We modify the consensus-based auction algorithm (CBAA) so that it can dynamically reallocate tasks among autonomous vehicles that can flexibly find a path to reach multiple dynamic targets while avoiding unexpected obstacles and staying close as a group as possible simultaneously. We propose the core algorithms and simulate with many scenarios empirically to illustrate how the proposed framework works. Specifically, we show that how autonomous vehicles could reallocate the tasks among each other in finding dynamically changing paths while certain targets may appear and disappear during the movement mission. We also discuss some challenging problems as a future work.

**Keywords:** autonomous vehicles; consensus decision making; task reallocation; team formation; obstacle avoidance path planning; auction mechanism

## 1. Introduction

The global market of using autonomous vehicles has grown substantially in recent years and has become an important tool for commercial, government and consumer applications. It can support solutions in many fields and is widely used in construction, oil, natural gas, energy, agriculture, military and other fields. Autonomous Vehicle applications have expanded from the traditional ground-based collection and delivery problem extends to air, underwater even to space applications. Potential applications for autonomous vehicle systems include space-based interferometers, military mission execution [1] (i.e., combat, surveillance and reconnaissance systems), hazardous material handling, distributed re-configurable sensor networks [2]. The operation of autonomous vehicle has also been advanced from single vehicle to multi-vehicle systems in the field. The coordination between autonomous vehicles becomes a challenging issue for multi-vehicle systems during operation. In the autonomous vehicle operations, there are tasks in controlling the movement situation such as dynamic path planning, mission planning, multiple obstacle avoidance and task coordination among vehicles in response to the state and environmental changes. These tasks become more complex and interesting since the dynamic and unknown environment can make the autonomous coordination among vehicles even more demanding and challenging to achieve.

In this work, we treat the autonomous multi-vehicle team as a Multi-agent System (MAS) and thus the terms "autonomous vehicle" and "agent" may be used interchangeably

without special mention in many situations of the paper. An agent could be a vehicle, a robot or an UAV operating in a 2D situation. A MAS is a system that adopts multiple interacting intelligent agents to simulate distributed decision making as a coordinated group to solve a complex problem. MAS becomes popular and important because of the following advantages but not limited to:

1.  Easy to scale up to deal with complex problems: A single agent can only execute a limited function and task. In a more complicated situation with multiple tasks, MAS is more suitable to fulfill the required demands.
2.  Increase the efficiency: The required task can be completed faster with MAS than a single agent system because multiple agents can solve several tasks in parallel rather than execute the tasks in sequence.
3.  Improve the reliability: A malfunction of a single agent system results from the mission delay or failure. However, a MAS executes tasks cooperatively as a group. The effective collaboration and synchronization of multiple agents can construct a more reliable system even when partial agents fail.
4.  Save the cost: It is cheaper and easier to implement a batch of simple systems than to build a complicated multi-functional system. It is in a divide-and-conquer strategy than implementing a closely coupled complex system. The maintenance cost is also much less.

However, there are still many challenges in building an applicable and robust MAS for real applications, particularly in autonomous multi-vehicle problems. The challenges include task allocation, group formation, cooperative object detection and tracking, path and trajectory planning, collision avoidance and much more in a complicated dynamic environment and task requirements [3]. For example, Thibbotuwawa et al. [4] point out unpredictable weather and energy consumption will post challenges to the routing and scheduling of UAVs to deliver goods from a pot to customer locations. In the study of Jones et al. [5], time for robotic planning (path planning, task planning, and mission planning) during SWAT action is an critical factor for the mission. Based on the structure of decision-making, an autonomous multi-vehicle system can be categorized from centralized to distributed [6]. In this paper, we model our system as a distributed MAS which does not require a central decision maker. Each agent can make a local optimal decision based on its own environment and goals. It does not rely on complete global information across the entire domain. However, each agent can make an optimal decision and action based on local information and communication with neighbors and achieve collective behaviors that benefit as a whole. It has the advantages of high stability and flexibility. In addition, when the regional decision-making exchange information is completed based on their respective local information, a rule that can achieve internal consensus decision-making is required to obtain the local optimal solution. This is also a topic that needs to be addressed in decentralized decision-making, i.e., how to achieve the best communication and cost efficiency with limited resources in a complex and unknown environment.

It is a challenge to avoid collisions among each other, especially when performing multi-agent cooperative tasks. They must also be able to dodge the obstacles in the external environment automatically. A complete set of movement path simulation software will need to integrate with the movement map to include natural landscape terrain, information of buildings and major objects [7].

The automatic task allocation problem is one of the challenging problems of a multi-agent system as well. Task allocation is defined as assigning several tasks to several agents and the agents have to carry out those assigned tasks. It is a general form of assignment problem, which belongs to fundamental combination optimization in mathematics. Each agent may have different capabilities and cost to perform different tasks. Each task must be assigned to at least one agent to be carried out. The cost of the problem depends highly on the outcome of the agent-task assignment. It cannot easily reach a consensus to an optimal cost solution within feasible time-bound by a team of fast-moving agents.

Many novel computer algorithms have developed in recent years to deal with the above problems. The traditional path planning problem combined with the application of multi-agent has also been updated in many new algorithms, such as artificial immune algorithm [8], genetic algorithm [9], random algorithm [10] and other combinations. Deep learning [11], flock of birds or a school of fish algorithm [12], dynamic windows approach [13], Consensus-Base Auction Algorithm (CBAA) [14] and Consensus-Based Bundle Algorithm (CBBA) [15] are dedicated to path optimization multi-agent obstacle avoidance and collaborative task allocation comprehensive algorithms. The research clusters can mainly be divided into two types: centralized and decentralized. Each has its own advantages and disadvantages. The former requires a leader (decision-making) and a centralized system to collect all evidence to complete the final decision. The advantages are faster decision-making and easier to get the best solution in the whole domain. The disadvantage is that once the number of groups and external variation are large and the amount of information is large, the decision-making speed and efficiency may slow down. Furthermore, once a problem occurs in the centralized system network or the decision-maker suffers in the event of an intrusion or attack, the followers or the entire system may break down.

In this research, we proposed a method of using distributed multi-agent systems to achieve the autonomous path planning, obstacle avoidance and dynamic task allocation with minimum cost and maximal reward. The proposed method allows the multi-agent system to bid task (target) and thus divide into subgroups and then plan a trajectory to avoid obstacles autonomously. This is especially important for military tactical and strategic applications. Multi-agent systems need the ability to process the collected data, coordinate via communication, make decision autonomously based on the assigned missions and changing environmental situations. This requires the agents to reach agreements and make decisions in real-time [16]. One important factor in increasing efficiency and effectiveness is to group agents into several compacted subgroups. The compacted group can avoid detection by the enemy and increase the information security during the movement mission. Therefore, we add compactness as one of the requirements into the obstacle avoidance algorithm in terms of cost. In summary, the overall goals of our multi-agent system are [17]:

1. Dynamically maintaining compact formations and adaptive task allocation by consensus protocols among agents;
2. Planning a dynamic path in a complex unknown environment;
3. Forming reconfiguration flexibly while mission changes or team members change their behaviors.

There are two major phases in our algorithm to achieve the above goals. In phase one, the algorithm is designed to assign each task to multi-agent by Committee and Consensus Base Auction Algorithm (C-CBAA). The second phase of the algorithm aims to avoid obstacles and plan trajectory autonomously. We call this Committee-Based Consensus Dynamic Trajectory Planning Algorithm (C-CDTP).

In Section 2, we brief some related works on task allocation and path planning. We define the objectives and describe the proposed algorithms in Section 3. We then set up different scenarios and show the experimental results in Section 4. In Section 5, we conduct analysis and discussion on the results. In Section 6, we make a conclusion and point out future work.

## 2. Related Work

### 2.1. Task Allocation

In a conventional task allocation problem with a group of $i$ agents and a set of $j$ tasks, each agent can execute a single task and each task can only be assigned once. The process costs and rewards depend on the kind of task and agent. The goal is to minimize costs with the maximum reward and efficiency. Recently, numerous methods have been developed to resolve the multi-agent task allocation problem with either the centralized or distributed algorithm. In distributed multi-agent task allocation problem, the task allocation algorithm can affect the overall performance as one of the leading research topics. The task allocation

problem is a dynamic decision-making problem because it depends on time and changes of environment and/or mission requirements [18]. Adding specific requirements will also increase the complexity of the task allocation problem [19]. Many solutions had been proposed to use a smart multi-agent system for task allocation. These solutions focused on two topics. The first is to assign a set of tasks to a group of agents successfully. Lemaire et al. [20] shows an incremental task allocation algorithm based on the Contract-Net protocol. They proposed a parameter in order to balance the workload between the different robots and to control the triggering of the auction process. Second, agents formed by operators need to reach an optimal and robust consensus for mission requirements. In 2008, Han-Lim Choi et al. [14] and Brunet et al. [21] proposed the CBAA method with a market-based decision strategy to solve the distributed multi-assignment problem. In 2011, Matthew Argyle et al. [15] extended CBAA to a multi-team structure on the distributed multi-assignment problem and assigned a sequence of multiple tasks by one task to one UAV at a time. In 2014, Darren Smith et al. [22] proposed another extension method to reduce the number of communication volume to complete a task allocation process used in unmanned vehicles. Shuo et al. [23] applied this problem to the military to develop the task allocation of wandering ammunition group and considered the unique attack constraints in the bomb combat mission.

Based on Argyle et al. [15], we propose a committee-based dynamic task allocation algorithm not only to keep CBAA's advantage of reaching an agreement within the distributed system but also improve it to allow each task to be assigned to autonomous multi-vehicle in the committee to reach consensus. In our autonomous vehicle system, we define the target as the task and a autonomous vehicle as an agent.

### 2.2. Path Planning

When used in autonomous vehicle systems, preventing inter-vehicle crashing and obstacle collision is the most critical issue when forming coordination [13]. In the previous papers, the main approaches can be seen as three categories: behavioral structure, leader–follower and virtual structure. Each of the methods has advantages and disadvantages.

The behavior-based structure aims to design different robot behaviors such as avoiding static obstacles, avoiding robots, moving to targets and maintaining formation [24,25]. The predetermined formations can also be switched by using graphical theory. The advantage of the structure is that it requires less communication with other robots due to decentralization. However, it is unlikely to deal with problems in a more complex environment and hard to prove whether the weight of each behavior is optimal computationally. Therefore, more machine learning researches have addressed this issue [26].

Tanner et al. [27], Fredslund and Mataric [28] and Luo et al. [29] proposed leader–follower approaches to deal with autonomous vehicles that have limited sensing scope. Each autonomous vehicle, excluding the leader, selects a neighbor to follow by a predetermined protocol. The graph of the neighbor relationship can be seen as a spanning tree, where each autonomous vehicle can maintain a related position to its neighbor. Thus, it is not required to sense every other autonomous vehicle's position. However, the formation shape is limited due to the leader–follower structure among neighbors.

All the autonomous vehicles have a geometric relationship based on a virtual point or virtual leader formations in the virtual structure [30]. Compare to the leader–follower approach, a virtual leader structure is proposed to improve its robustness since every autonomous vehicle will not directly affect others' direction but one obvious disadvantage is that reconfigure the formation is more challenging. Rezaee and Abdollahi [31] proposed a decentralized scheme based on both behavior and virtual leader structure to control the formation and avoid obstacles, respectively.

To solve the above problems of the three structures, the research about consensus theory has emerged as a challenging topic in recent years. Ren [32] has proved that behavior-based, leader–follower and virtual structure can be regarded as special cases of consensus theory. Therefore, many papers combined different formation control structures

by designing consensus protocols. Baranzadeh and Nazarzehi [33] used consensus variables calculated among a autonomous vehicle and its neighbors to stabilize the formation. Dong et al. [34] proposed a multiple-leaders scheme where all leaders should reach a consensus before configuring a formation and the followers then converge to the convex hull formed by those of leaders. However, the two methods are not able to deal with problems such as formation splitting and merging. Thus, Alonso et al. [35] and Zhu et al. [36] proposed a decentralized base formation control in a dynamic environment. With different predetermined formations, the team can reach a consensus when encountering obstacles or a narrow trail. A consensus protocol also determines formation splitting or merging. Min et al. [37] proposed a virtual leader approach, which only requires local knowledge given by a UAV's neighbors. Each UAV only needs to maintain the relative position to keep the formation. Amanatiadis et al. [38] discussed the constraints of a multi-tasks path planning problem for a single robot in an unexplored environment. They proposed a Cognitive-based Adaptive Optimization (CAO) algorithm to complete operation with minimum time.

Although many researchers have investigated the formation control problems, few works focus on reconfiguration issue or a relatively more complex environment of high uncertainty. Based on Min et al. [37], we propose a virtual center of gravity as a virtual leader that we can flexibly adjust the positions for every autonomous vehicle. Therefore, our work allows autonomous vehicle teams to avoid obstacles and inter-collision as the higher priority and maintain distances among teammates in all situations.

## 3. Methodology

Different cost and operation considerations in practice can induce different types of feasible solutions. Consider the multi-agent multi-task assignment problem where a group of $N_u$ agents to approach $N_t$ targets (tasks) while trying to maximize the reward of each agent. The objective of task allocation for a distributed agent system is to find an optimal solution with enough number of agents in the subgroup using the minimum cost of path planning. The minimum costs include the distance to the target, maintaining the team formation without inter-collision and avoiding obstacle collision. In this research, we propose the C-CBAA and C-CDTP for task allocation and path planning, respectively.

We consider a team of $N_u$ agents in an unknown environment, each of which has a range of sensing areas detected by its sensors. By knowing the positions of obstacles and their teammates in the sensing area, every agent should move from the starting area to the target under the constraints of avoiding obstacles and keeping a safe distance among teammates. Besides, a compact team structure needs to be figured during the whole mission in various testing environments.

### 3.1. Model Formulation

Each agent must face with many different scenarios and therefore can have complex internal states to cope with complicated tasks under different situations. At the high level of model formulation, we model each agent with three major internal states that are switching among each other to deal with different complicated situations. Figure 1 shows the three *moving*, *waiting* and *maneuvering* states and how a state changes to another under different situations. Most of the time, agents maintain at the *moving* state. Once the set of trajectory nodes are calculated and returned from the algorithm, the agent follows the nodes and simultaneously predicts a new trajectory. The loop is continued until another state is switched or the target is reached. In any case during the movement, if all of the predicted trajectories are blocked by the teammates or obstacles, the agent state will be automatically switched to the *waiting* state. The *waiting* state refers to an agent which stops and remaining at the same position to re-plan another path. It is usually occurs when avoiding obstacles, formation merging especially when the orientations change frequently. Thus, a later agent should give up its path priority until the former agent pass.

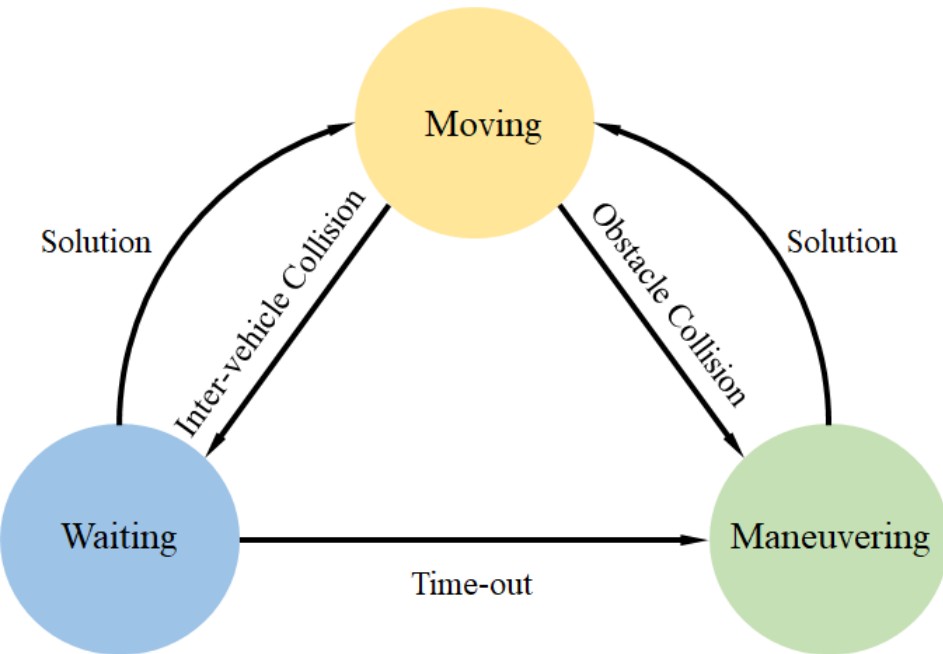

**Figure 1.** State switching under different circumstances.

However, a deadlock may occur especially when most agents are congested in such an environment as in a blind alley. To solve such a problem, we set a constant *clock* to prevent an agent from waiting for overtime. If *clock* exceeds the waiting time bound, a *maneuvering* state will be invoked, as shown in Equation (1).

$$waiting() = \begin{cases} \text{stop,} & \text{if } clock < t_{wait} \\ \text{switch to } maneuvering, & \text{otherwise} \end{cases} \tag{1}$$

In *maneuvering* state, the agent stops and rotates its orientation in search for other possible trajectories to the target. The *maneuvering* state is also invoked in other cases such as facing with multiple or irregular-shaped obstacles. We implement the agent in *maneuvering* state to rotate clockwise along the orientation with a predefined angular velocity $\omega_{nav}$, that is calculated according to Equation (2).

$$\begin{aligned} v_i &= 0 \\ \theta_i(t) &= \theta_i(t_0) + t \cdot \omega_{nav} \end{aligned} \tag{2}$$

By changing the orientation of the agent by rotating, it is possible for agent to find other alternative paths when the leading agents are facing with deadlock situations or current feasible paths are blocked by all team agents. Furthermore, even waiting cannot expect to find a feasible path efficiently to resolve the deadlock.

The starting point and the target (the destination point) of each agent are given initially by

$$\begin{aligned} Start_i &= x_i(t_{start}), y_i(t_{start}) \\ Target &= x_{end}, y_{end} \end{aligned} \tag{3}$$

where $x_i(t_{start}), y_i(t_{start})$ is the position of $i$th agent at the starting time $t_{start}$. We then use the linear discrete-time model $(t_0, t_1, ..., t_n)$ for the kinematic equations as in Equations (4).

$$x_i(t_n) = x_i(t_0) + \int_{t_0}^{t_n} v_i(t) \cdot cos\theta_i(t)dt$$

$$y_i(t_n) = y_i(t_0) + \int_{t_0}^{t_n} v_i(t) \cdot sin\theta_i(t)dt \tag{4}$$

$$\theta_i(t) = \theta_i(t_0) + \int_{t_0}^{t} \omega_i(\hat{t})dt$$

for all $i \in \{1, 2, ..., N\}$, where $(x_i(t_n), y_i(t_n))$ is the position of $i$th agent, $v_i(t)$ is the velocity and $\theta_i(t)$ is the orientation at time $t$. $\theta(t)$ is determined by the angular velocity $\omega$ at time $\hat{t} \in [t_0, t_n]$. In other words, $v$ and $\omega$ are two inputs which control the motion of agents at any time. In practice, the agents have the speed and angular velocity constraints as shown in Equation (5).

$$0 \le v_i(t) \le v_{\max}$$

$$- \omega_{\max} \le \omega_i(t) \le \omega_{\max} \tag{5}$$

On a battlefield with high uncertainty, we use the dynamic windows approach for each agent to predict $j$ trajectories $T_i^j$ in a time window $[t_0, t_T]$. Each trajectory consists a series of node coordinates from time $t_0$ to time $t_T$ as shown in Equation (6) and we define the last node of $T_i^j$ at time $t_T$ as the allocated target point $P_i^j$.

$$T_i^j = \{(x_i^j(t_0), y_i^j(t_0)), (x_i^j(t_1), y_i^j(t_1)), ...P_i^j\}$$

$$P_i^j = x_i^j(t_T), y_i^j(t_T) \tag{6}$$

The above predicted possible trajectories depend on the inputs $v$ and $\omega$ controlled by time windows and the range of $v$ and $\omega$ can be adjusted more tightly from Equation (7) as follows:

$$\max(0, \hat{v} - a \cdot \hat{t}) \le v \le \min(v_{max}, \hat{v} + a \cdot \hat{t})$$

$$\max(0, \hat{\omega} - \Delta\omega \cdot \hat{t}) \le \omega \le \min(\omega_{max}, \hat{\omega} + \Delta\omega \cdot \hat{t}) \tag{7}$$

where constant $a$ is the velocity acceleration and constant $\Delta\omega$ is the angular acceleration. Each agent predicted several possible trajectories then select the best trajectory as the final trajectory at time $t_0$ by using the consensus scoring system to be discussed in the next section.

### 3.2. Consensus Scoring System

In the distributed architecture, each agent's behavior depends not only on its own decision but also on the behaviors of other teammates and the environment. However, Equation (8) shows that the visual scope of each agent is limited by its sensor, in which we define a set of neighbors as virtual committee members for each agent. Let $A_i$ denote the sensing area of $i$-th agent with radius $r_i \in \mathbb{N}$, where $r_i$ should always be greater than its safety distance $d_i \in \mathbb{N}$. $d_{nearest}$ is the distance of agent $i$ and its neighbor $k$, $N_i$ is the set of the neighbors of $i$-th agent. We then define the teammates who can be sensed in the area as the neighbors of $k$-th agent. Since the sensing area and the safety distance of each agent may differ, the neighbor relationship between any two agents is essentially asymmetric.

$$d_i \le d_{nearest} \le r_i$$

$$d_{nearest} = min(\sqrt{(x_i - x_k)^2 + (y_i - y_k)^2}), \forall k \in N_i \tag{8}$$

$$N_i = \{N_i^1, N_i^2, ...N_i^n\}, \forall N_i \in A_i$$

We propose a scoring system to find the minimum-cost trajectory among the candidate trajectories predicted in the time window. The cost function of a candidate trajectory $j$ of $i$-th agent can be denoted in Equations (9):

$$Cost(i,j) = (1 - \alpha_i - \beta_i) \cdot Cost_{speed}(i,j) + \alpha_i \cdot Cost_{form}(i,j) + \beta_i \cdot Cost_{target}(i,j)$$
$$\alpha_i = 1 - \frac{d_{nearest}}{r_i} \tag{9}$$

where $Cost_{speed}(i,j)$, $Cost_{form}(i,j)$ and $Cost_{target}(i,j)$ are functions of speed cost, formation cost and path cost to the target of $i$-th agent's trajectory $j$, respectively. $\alpha_i$ and $\beta_i$ are respective the relative weight coefficients of formation and path cost to compromise these factors and find the best trajectory. During the agent mission, the co-operated agents do not only move toward the immediate target point but also move closer to the virtual center of gravity of its team. To evaluate this cost function, $Cost_{form}(i,j)$ is calculate according to the steps as shown in Equations (10)–(13), respectively. Firstly, each agent calculates a virtual center of gravity $G_i=(x_{G_i}, y_{G_i})$ based on the position of its team neighbors and itself along a candidate trajectory $j$.

$$x_{G_i} = \frac{1}{n+1} \cdot (x_i + \sum_{j \in N_i} x_j)$$
$$y_{G_i} = \frac{1}{n+1} \cdot (y_i + \sum_{j \in N_i} y_j) \tag{10}$$

Secondly, we predict the new center of gravity $PG_i$ after $t_T$ time window which is used to evaluate the best trajectory. $PG_i$ is calculated based on $G_i$ with the current speed and orientation.

$$x_{PG_i} = x_{G_i} + t_T \cdot v_i \cdot cos\theta_i$$
$$y_{PG_i} = y_{G_i} + t_T \cdot v_i \cdot cos\theta_i \tag{11}$$

Lastly, the overall formation cost and target cost functions after a time window are shown in Equations (12) and (13), respectively.

$$cost_{form}(i,j) = dist(P_i^j, PG_i) \tag{12}$$

$$cost_{target}(i,j) = dist(P_i^j, Target) \tag{13}$$

Furthermore, the scoring system try to achieve a balance between individual target-path cost and team formation cost. In general, the farther the distances between an agent with its target and its team, the faster it should move. To accommodate this scenario, the speed cost function $Cost_{speed}(i,j)$ is defined as the difference between the maximum speed and the speed at the predicted target point after the time window along candidate trajectory $j$ of $i$-th agent. The equation used is shown as Equation (14):

$$Cost_{speed}(i,j) = V_{max} - v_i \tag{14}$$

The definition indicates that if an agent $i$ moves at its maximum velocity along its trajectory $j$ the $Cost_{speed}(i,j)$ is zero.

### 3.3. Dynamic Task Allocation

The goal of our dynamic task allocation algorithm C-CBAA is to find a conflict-free matching of tasks to agents that minimizes the local cost to achieve the global reward. In our research, an assignment is said to be conflict-free if each agent is assigned to only one target but each target can be performed by multi-agent at the same time. In order to meet the needs of the actual goal, we proposed the idea of performing targets together by a team. This is the most different from the conventional task allocation concept.

Consider the agents and dynamic task allocation problem where a group of $N_u$ agents are planning to reach targets $N_t$ while they also manage to minimize the overall trajectory cost of the group as Cost. This can be stated formally as:

$$Cost = min \sum_{i=1}^{N_u} \sum_{j=1}^{N_t} Cost(i,j) T_i^j \qquad (15)$$

subject to

$$\sum_{i=1}^{N_a} a_{it} \geq G_t \qquad (16)$$

where $T_i^j$ means the trajectory $j$ of agent $i$ as defined in Equation (6). In Equation (16), $a_{it}$ indicates agent $i$ selects target $t$ and $G_t$ means the minimum number of agents that is required by each target $t$. In other word, a target requires a minimum number of agents to reach the target to be sufficiently carried out the task. If the total number of agents of the group cannot execute all task requirements of targets, the farthest target away from the center of gravity of the group will be dropped automatically. This is to ensure some targets can be carried out by some agents successfully rather than fail all missions due to the limited number of agents. Figure 2 shows the case of three agents in an operation of five targets. The horizontal and vertical axes represent the x and y coordinates on the ground by assuming agents start near the origin. The colored circles are agents in operation and the dash lines represent the paths to the assigned triangular targets. Black dots are random obstacles in the scenario. Each target requires one agent to execute the task, the farthest yellow and green targets are abandoned.

Our proposed automatic task allocation method is implemented in two phases: the auction phase and the consensus-conflict resolution phase. The former focuses on bidding a self-target and updating information, the latter is subjected to team communication and gain consensus to find the actual best agent to win in the auction process. Each iteration of the task allocation requires completing both phases in sequence to reach the consensus among agents in the task allocation.

Algorithm 1 shows the auction process of agent $i$ for target $t$ at iteration $\tau$. The re-auction process starts when there is a change in the number of targets. The conventional CBAA algorithm adds all agents to participate in the re-auction and award target $t$ to the agent with the highest price even if an agent has already been assigned to some target. This can cause redundant and inefficiency of task assignments. In order to improve the efficiency of CBAA algorithm, our C-CBAA algorithm as shown in Algorithm 1 only allows partial agents to re-bid the un-allocated targets as shown in Equation (17).

$$h_{it} = \amalg(c_{it} < Cost \,\&\, z_{it} > 0) \qquad (17)$$

$\amalg(\cdot)$ is a binary operation judgment on whether the condition is true or false, $c_{it}$ is the agent $i$'s minimal trajectory cost for target $t$ using a condition defined in Equations (15), $z_{it}$ means agent $i$ stores the least number of agents required for target $t$. This condition reduces the number of participants in the bidding process and the number of auctions which are already awarded. Therefore, only when a target disappears or the number of agents assigned becomes less than the number of agents required by the changed target, an auction bidding process is invoked. The performance comparison is presented in Section 4.2.2.

Algorithm 2 is the consensus and conflict resolve phase. Agent $i$ will receive the information from all its neighbors $k$ in each iteration. The agent $i$ will determine if there is a conflict between the target $T_i$ selected from the previous phase and other agents conduct the re-bidding process. Finally, if the agent and its neighbor(s) are assigned to the same target, it will add as the neighbor(s) to its teammate.

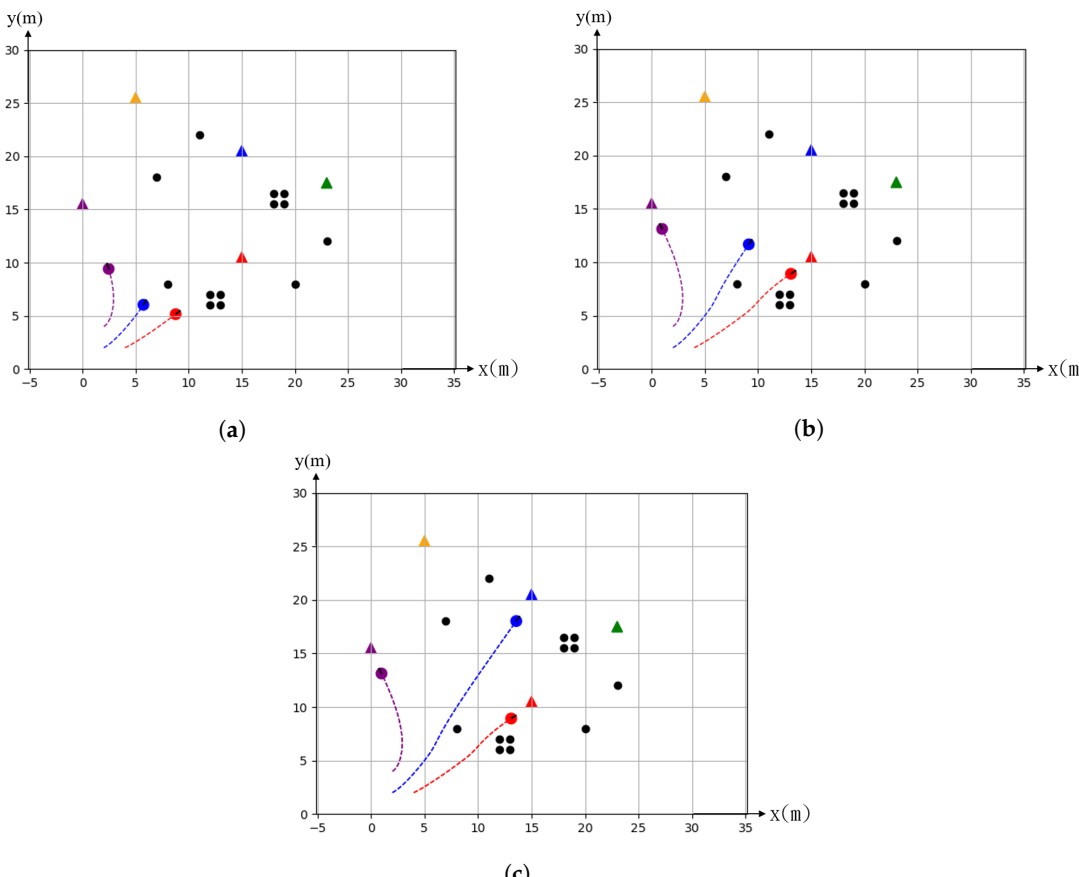

**Figure 2.** Simulation of scenario when number of targets exceeds the number of agents: (**a**) Initial state: 5 targets and 3 agents, (**b**) Midway state: agents search target and (**c**) Final state: agents arrive at their assigned targets and abandon the farthest targets away from the center of the global gravity.

---

**Algorithm 1** C-CBAA *Phase* 1 for agent *i* at iteration $\tau$

---

 1: Update the previous iteration data of agent *i* to the current data
 2: procedure DETECT TARGET
 3: **if** Discover new targets **then**
 4:      re-allocate $z_{it}$
 5: **end if**
 6: **if** target disappeared **then**
 7:      Remove disappeared target info from agent *i*
 8:      re-allocate $z_{it}$
 9: **end if**
10: procedure AUCTION
11: **if** UAV *i* does not allocate any target **then**
12:      $h_{it} = \amalg (c_{it} < Cost \,\& \, z_{it} > 0), \, \forall \, t \in T$
13:      **if** $h_{it}$ is True **then**
14:           Unallocated targets assigned to agent *i* with the minimize cost.
15:           update $z_{it}$
16:      **end if**
17: **end if**
18: **if** UAV *i* needs to re-bid a new target **then**
19:      Compare the quantity difference between agent *i*'s teammates *and* $z_{it}$
20:      Decide whether the agent *i* will re-bidding
21: **end if**

---

---

**Algorithm 2** C-CBAA *Phase* 2 for agent *i* at iteration $\tau$

---

1: EXCHANGE info between neighbors
2: procedure CONSENSUS and CONFLICT RESOLUTION
3: Update information from neighbors
4: Calculate $c_{it}$ with updated information
5: **if** UAV *i* did not award the target **then**
6:     UAV *i* clear all the information, update $z_{it}$ and re-bidding
7: **end if**
8: procedure UPDATE COMMITTEE
9: agent *i* exchange information with neighbors.
10: **if** agent *i* and its neighbor(s) are assigned to the *same target* **then**
11:     Add the neighbour(s) to its teammate
12: **end if**

---

### 3.4. Distributed Dynamic Path Planning Algorithm

Our proposed C-CDTP algorithm is illustrated in Algorithm 3. Firstly, each agent predicts trajectories based on a time window, rejecting the ones which may collide into obstacles or teammates. After this step, we can assure that obstacles and inter-collision will not occur at any time (lines 4–20). However, we should identify the difference between the two circumstances by using a flag *interCollision* which may affect the agent's decision to change its state at the end of the algorithm. Secondly, the agent finds the minimum-cost trajectory by using the scoring system. Lastly, the agent decides its behavior among the three states: moving, waiting and maneuvering. In the *maneuvering* state, the agent starts rotating in search of feasible moving direction to reach the target.

---

**Algorithm 3** Dynamic trajectory planning

---

1: $cost_{min} = \infty$, $traj* = [\ ]$
2: $interCollision = false$
3: Calculate dynamic windows
4: **for** $v$ in time window **do**
5:     **for** $\omega$ in time window **do**
6:         predict a possible trajectory *traj*
7:         **if** *traj* may collide into obstacles **then**
8:             continue
9:         **end if**
10:        **if** *traj* may collide into teamates **then**
11:            *interCollision = true*
12:            continue
13:        **end if**
14:        calculate *cost* by using the scoring system
15:        **if** $cost_{min} > cost$ **then**
16:            $cost_{min} = cost$
17:            $traj* = traj$
18:        **end if**
19:    **end for**
20: **end for**
21: **if** $traj*$ exists **then**
22:     **return** $traj*$
23:     **if** *interCollision* is *true* **then**
24:         **return** *state = waiting*
25:     **end if**
26: **end if**
27: **return** *state = maneuvering*

---

## 4. Simulation Results

### 4.1. Scenarios

Supposed we have a mission of using agents to detect the enemy command center and attack potential ground targets in combat. Agents need to maintain at low altitude movement to avoid obstacles, search targets and avoid radar detection. We can have the following three possible scenarios:

1. Single target scenario: A single task is assigned to a group of agents, for example, to survey the enemy command center. Agents must plan the path to the target and conduct comprehensive surveillance.

2. Multiple static targets scenario: The mission is to detect, identify and attack multiple targets. For example, a missile defense system includes a radar system, command and control center, communication system and missile launchers [39]. A fleet of agents are taking off from the same site must be able to use the acquired knowledge to evaluate and group to smaller squads to execute their mission independently.

3. Multiple dynamic targets scenario: Number of targets may increase or decrease during the operation before all agents reach their destinations.

   - Decrease target scenario: Targets may be canceled or disappear during operation. For example, a fleet of agents was launched to survey and attack an anti-missile defense system which includes the ground radars, interceptors, command and control centers. Agents take off from the same base and are grouped into four subgroups to attack individual targets. During the movement, the enemy command center may move to a new location away from the mission range. The number of targets decreased to three. It is also prioritized to destroy the ground radar system to disable the enemy's detection capability.

   - Increase target scenario: New targets may also appear during the operation. For example, the initial mission is to survey the enemy command center. The enemy adds two interceptor launchers within the operation area before agents reach the target. Thus, the target number increases from one to three (command center and two interceptor launchers).

### 4.2. Experiments

We implemented the autonomous agents system with Python tool kit on a PC with Intel (R) Core(TM) i7-8550 running at 1.8 and 1.99 GHz and used simulation to present the results of task assignment of agents with our C-CBAA method and Committee-Based Consensus Dynamic Trajectory Planning algorithm to show their 2D path with different scenarios. Four cases were simulated to demonstrate how agents can coordinate task assignment and path planning autonomously. Section 4.2.1 shows the flexibility and efficiency of our trajectory planning compression by passing through a complex environment. Section 4.2.2 focuses on the case of static target scenario. Section 4.2.3 shows the result of assignment change during the movement by either increasing or decreasing the number of targets. Section 4.2.4 compares the differences between the proposed C-CBAA method with the traditional CBAA algorithm. The parameters used in our simulation are shown in Table 1.

The followings are the simulation results of multi-agents in different scenarios. The dash lines are the flight trajectories of agents. Agents are colored dots and different color represents different subgroup. The triangle symbols are the targets. Greyed areas and black dots represent the obstacles.

**Table 1.** Simulation parameters.

| Parameter | Value |
|---|---|
| UAV radius ($r$) | 0.4 m |
| safe distance ($d_{safe}$) | 0.8 m |
| minimum velocity ($v_{min}$) | 0 m/s |
| maximum velocity ($v_{max}$) | 15 m/s |
| maximum acceleration ($a_{max}$) | 0.3 m/s$^2$ |
| minimum yaw rate ($\omega_{min}$) | $-40 \cdot \frac{\pi}{180}$ rad/s |
| maximum yaw rate ($\omega_{max}$) | $40 \cdot \frac{\pi}{180}$ rad/s |
| maximum yaw acceleration ($\omega_{max}$) | $40 \cdot \frac{\pi}{180}$ rad/s |
| predicted time ($t_T$) | 2 s |
| time tick | 0.2 s |

4.2.1. Passing through Complex Environment

To evaluate the capability and flexibility of the configuration formation change in terms of compactness, we set up a narrow passage for the teams to pass through. In the simulation, agents are expected to change their formation by switching their *moving* states to *waiting* states or vice versa in order to orderly pass through the narrow passage. The result is shown in Figure 3 where a team of 8 agents start from the left and the target is set on the right. We use the color for agents to identify the states while passing through the passage, Figure 3a is evident that both sides of the agents predict the inter-collision may occur before entering and therefore they wait for the other teammates to pass first. In Figure 3b where the green agents represent the ones in a *moving* state and the black ones are for the *waiting* state.

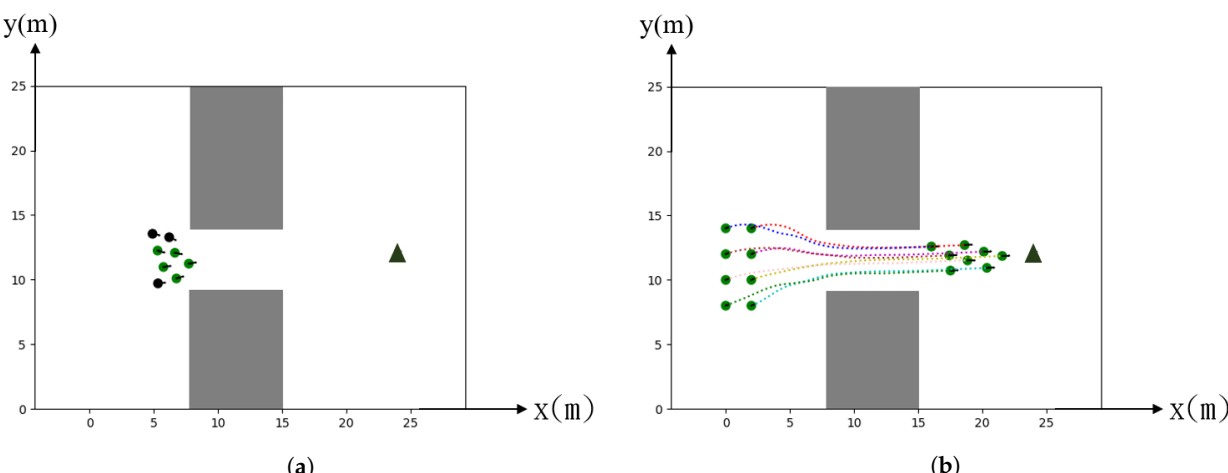

(**a**)   (**b**)

**Figure 3.** Simulation of trajectories in a narrow passage: (**a**) Waiting state validation, (**b**) All passing.

More complex obstacles are designed to evaluate the comprehensive functionality of agents in changing formation and finding a path to get around the obstacles. Due to the range limitation of sensors, agents cannot predict the obstacles that may block their moving direction far away in advance. Our algorithm requires neither fixed leader nor fixed formation so that the team can easily move in any direction even the opposite one in order to find an alternative path solution in facing with an environment with complex distribution of obstacles.

Figure 4a shows that the 6 agents are encountered with a dead-end corner shortly after they moved for some distance at the beginning. Nevertheless, the leading agents would invoke a *maneuvering* state when they found that no path was available. On the other hand, the following agents behind would switch the *moving* state to the *waiting* state and thus waited for their teammates to move. To prevent the deadlock for the whole team to wait for each other, we preset a time-out variable *clock*. The *clock* is randomly assigned from 2 to 3 s for dealing with the synchronizing problem. That is, every agent will start to move and search for new feasible paths after the time-out. Finally, if there is a feasible path, the team would successfully find the alternative route and then continue to move toward the target.

Figure 4b shows the analysis of agent's performance from $N = 1$ to 15, where $N = 1$ represents a single agent which does not require formation coordination. The green line denotes the agent's average distance, while the red line denotes the total spent time. Since we define $v_{max} = 1$ m/s in the simulation, the average waiting time when agents are being stuck can be analyzed from the difference between both lines. It can be seen that the average distance is not affected much by increasing the number of agents that implies the stability and efficiency of the formation coordination algorithm. On the other hand, the total time spent depends on $N$ due to the waiting of agents to get around the obstacle that blocks the team moving direction. In the situations where agents attempt to find a way out of the obstacles, each agent will switch between the *waiting* and the *maneuvering* states to prevent a deadlock. In another word, its behavior is somewhat similar to data retrieval in a stack (i.e., last-in-first-out structure) and the total time spent is a linear growth depends on $N$.

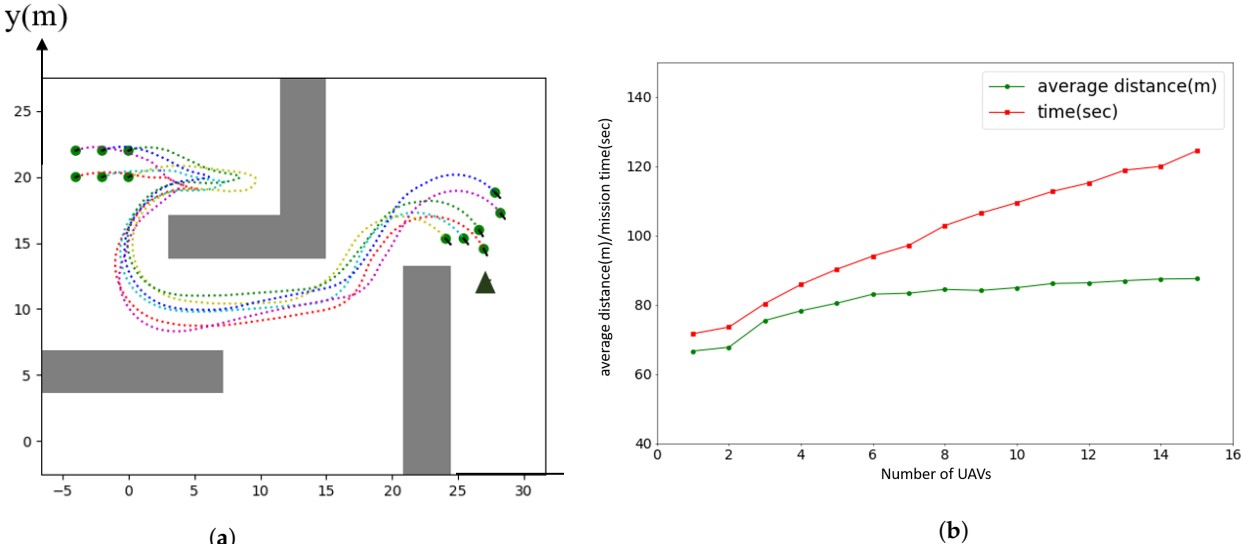

(**a**)

(**b**)

**Figure 4.** UAVs in Maze-like environment and Analysis: (**a**) All trajectories in a maze-like environment, (**b**) Line chart of the average distance and the total time spent.

4.2.2. Static Target

Figure 5 shows the simulation of static single target. We place random obstacles between the starting and target positions to evaluate the coordination capability between agents using our algorithm. The number of agents as well as the number and distribution of the obstacles can be given arbitrarily. Figure 5a shows agents launch around the starting positions near the lower left corner toward the target position near the upper right corner. Agents must achieve the goal position by adjusting their paths and maintaining as a compact group as possible to avoid obstacles. The agents do not know the obstacles prior to realizing that they get near an obstacle or other agent by a sensor. The obstacle avoidance path planning is conducted autonomously by agents to minimize the cost. Agents can adjust their trajectory constantly to avoid collision between agents and environmental obstacles. Agents can aggregate as a compact group afterwards and fly to the destination as shown in Figure 5b. We run the simulation many times with random distribution of obstacles to evaluate the robustness of path planning. Obstacle avoidance and all results turn out to be able to find a satisfactory trajectory to reach the static single goal. Each agent only needs to exchange information with its neighbor teammate because we design the algorithm in a distributed method in computing the path for each agent during the movement.

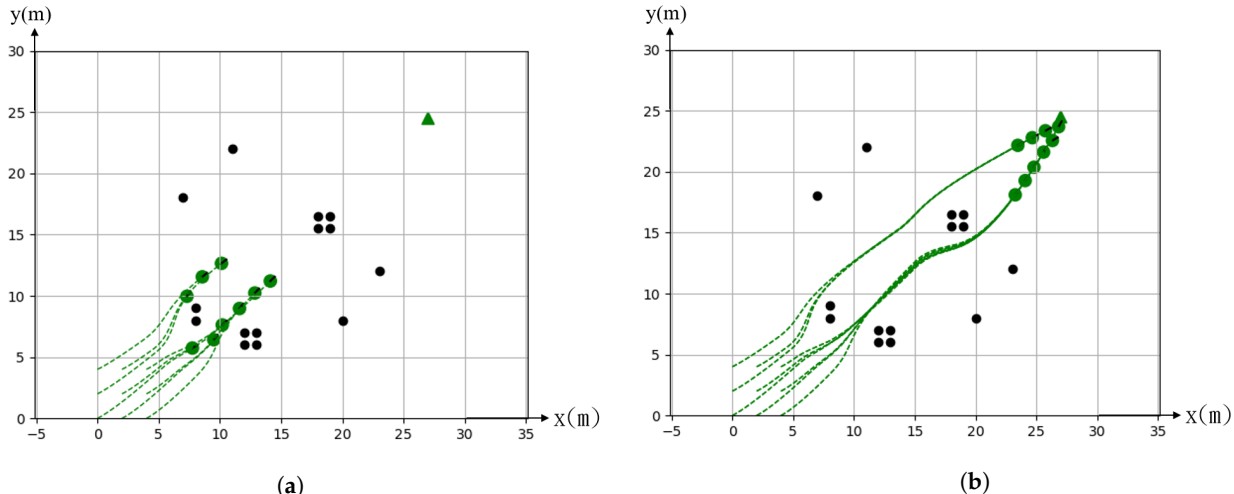

(**a**)  (**b**)

**Figure 5.** Simulation of static single target. (**a**) Original toward to the target. (**b**) Finally stop at the target.

Figure 6 shows the simulation of multiple static targets. The purpose of this simulation is to evaluate the effectiveness of our algorithm in the case of multiple targets and obstacles. In the experiments, we adopted nine agents that were randomly placed around the starting position and took off at the same time to approach four goal targets which are indicated by triangular symbols with a different color in a 2D space. Obstacles were placed randomly between the starting positions of agents and the targets with varying sizes, as shown in Figure 6a,b shows the result of the simulation using the proposed method. Agents were grouped into four subgroups and planned the trajectories autonomously. The trajectory and formation are based on minimum costs which include the distance, the compactness of the group without the inter-collision among agents and the collision with obstacles.

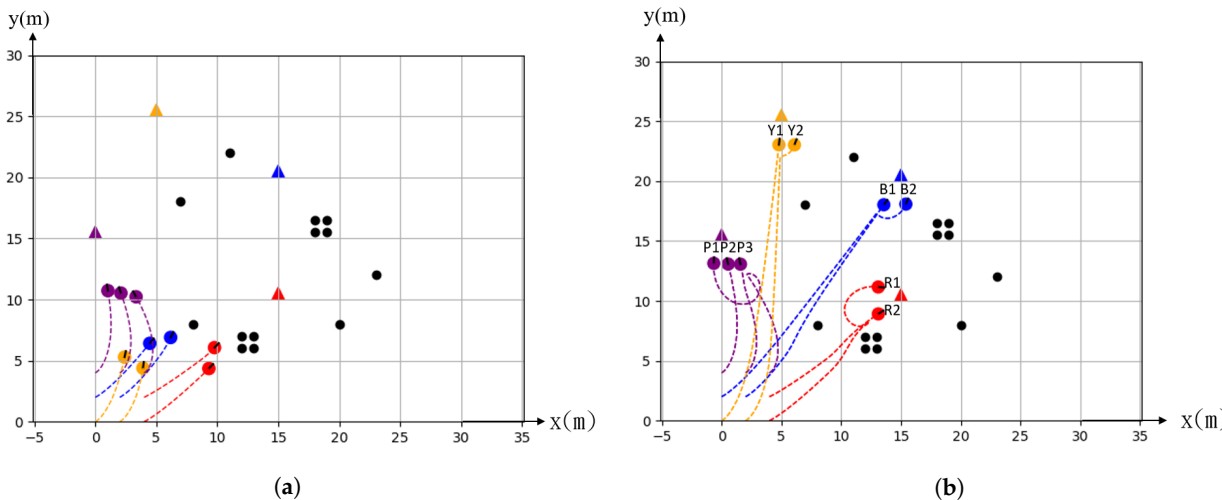

**Figure 6.** Simulation of static multi target. (**a**) Original toward to the target. (**b**) Finally stop at the target.

### 4.2.3. Dynamic Target

Since in a dynamic situation, the targets can appear and disappear, in order to evaluate the difference of the effects on our methods, we separate the experiments into two aspects, dynamically increasing the targets versus decreasing the targets. Figure 7 shows the simulation of increasing targets. The original targets are denoted by red, yellow and blue triangles. Nine agents launch from the starting positions and are demanded to approach the targets as groups Figure 7a. After 40 iterations of time steps, a new target (indicated as a purple triangle) was added and it turns out that the two subgroups in blue and yellow were reconfigured into three subgroups. We assume the newly added target has the same priority as the original blue and yellow targets. Moreover, the subgroup targets in red did not change their path Figure 7b.

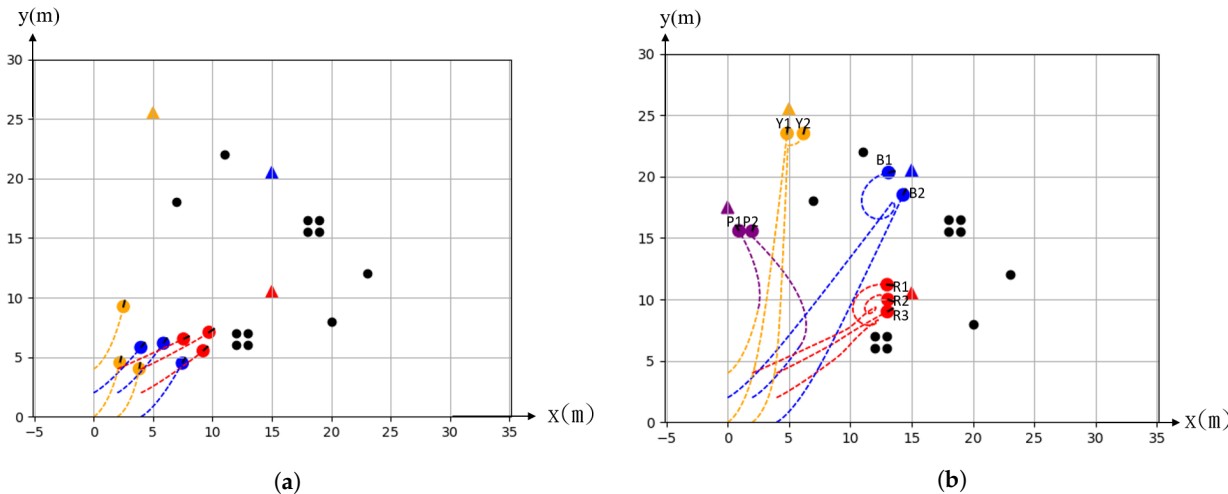

**Figure 7.** Simulation of increasing target. (**a**) Original toward to the target. (**b**) Finally stop at the each target.

Figure 8 target disappeared during the movement after 40 iterations of time steps. It turned out that the red group was split shows the simulation in decreasing a target. Four targets in terms of red, yellow, blue and purple colors and denoted with triangle symbols that appeared initially. A fleet of nine agent launched from the starting position and was demanded to move toward targets as four subgroups as shown in Figure 8a. We tentatively set that the red and joint to other groups that were approaching to the blue and yellow targets, respectively, as shown in Figure 8b.

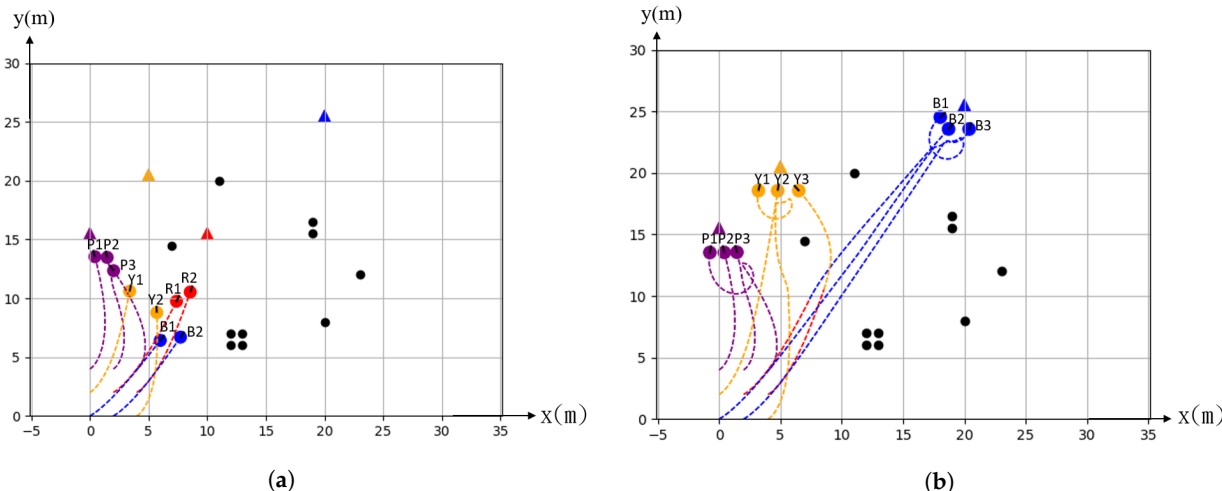

**Figure 8.** Simulation of decreasing target. (**a**) Original toward to the target. (**b**) Finally stop at each the target.

#### 4.2.4. Comparison with the Previous Method CBAA

In this section, we compared the performance between the CBAA and the proposed C-CBAA algorithm when the number of targets varies from one to three. Obstacles are also randomly distributed in the testing scenarios and the target(s) are randomly appeared and disappeared during the movement in the scenarios. The efficiency is based on the average number of iteration steps needed to finish re-allocation task due to the change of targets. We simulated the same scenarios with different number of agents (3, 6, 9, 12 and 15, respectively) and the results are as shown in Table 2 to compare the efficiency at situations with different agent complexity. Figure 9a shows the result of increasing targets while Figure 9b shows the result of decreasing targets. Both results show that the C-CBAA outperforms the CBAA algorithm against different numbers of agents. When the number of targets was increased randomly, C-CBAA converges 31% faster than the original CBAA on average. The advantage of using C-CBAA is more obvious in the case of decreasing targets. In Figure 9b, the red line shows C-CBAA converges 48.72% faster than CBAA. The reason is that the target reduction only needs to deal with the reduced target group. It merely re-allocates a target to each agent in the group again, as shown in Figure 9b. On the other hand, while the number of targets are increased, the groups responsible for other adjacent targets must be taken into consideration in order to re-allocate the new targets among them as shown in Figure 9b. It turns out that the performance of the C-CBAA algorithm takes less time to reach consensus than the CBAA when the number of targets is decreased. Furthermore, the difference of the average iterations is increased as the number of agents is increased. It implies that the C-CBAA is even more effective than the traditional CBAA algorithm in a highly complex mission scenario.

**Table 2.** Efficiency comparison between CBAA and C-CBAA algorithms under different target change scenarios.

| Conditions | Average Iterations ⟍ Agents ⟍⟍ Algorithms | 3 | 6 | 9 | 12 | 15 |
|---|---|---|---|---|---|---|
| increasing | CBAA | 2.55 | 4.7 | 6.2 | 6.9 | 8 |
| | C-CBAA | 1.5 | 3.5 | 4.25 | 4.8 | 5.5 |
| | average improvement(%) | 31% | | | | |
| decreasing | CBAA | 2.6 | 5 | 7.3 | 10 | 12.1 |
| | C-CBAA | 1.7 | 2.6 | 3.5 | 4.6 | 5.4 |
| | average improvement(%) | 48.72% | | | | |

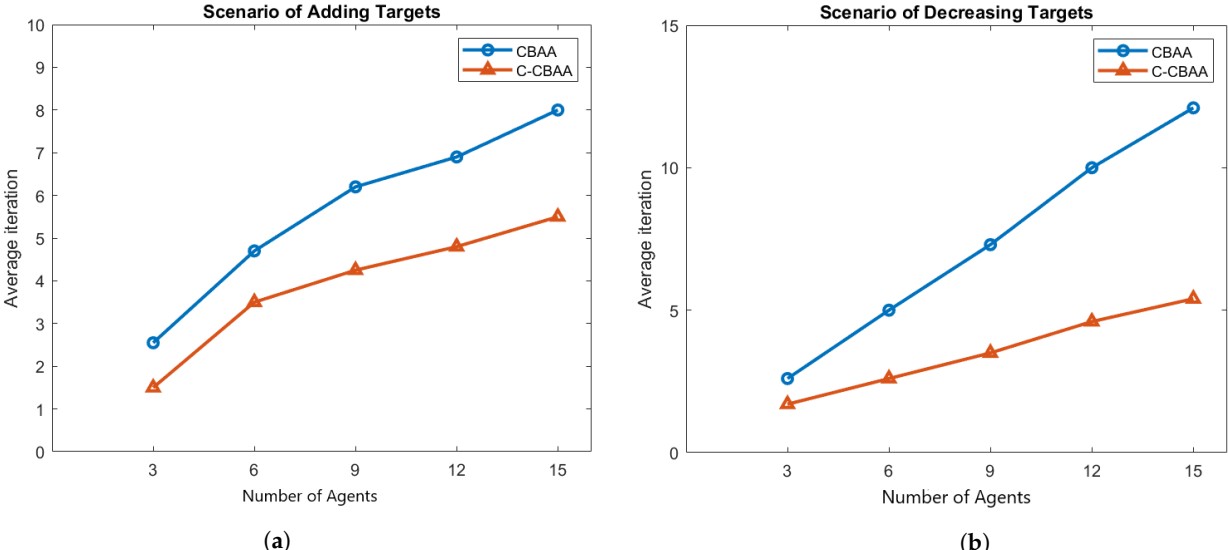

**Figure 9.** Compare the performance of two algorithms. (**a**) Increasing targets. (**b**) Decreasing targets.

## 5. Analysis and Discussion

The simulation results show that our algorithm can be applied to the operation of multi-agent systems for both single and multiple targets at either static or dynamic scenarios. The proposed C-CBAA algorithm has various advantages on path planning and task allocation in multi-agent mission operations:

1.  Collision avoidance between agents and obstacles: The C-CBAA use a lead-trail formation as the preferred configuration formation to avoid collisions among agents. As we can see from Figure 5, all agents in a group maintains a linear separation to avoid a collision before arriving at the target positions. Trail formation is also easier to go around the obstacles without interference by the other agents.

2.  Reach the targets simultaneously with a minimum cost: agents are grouped based on the initial target requirements such as the number of agents needed and the distance of each agent to its target. This ensures agents in the same group reach the assigned target at the same time to execute the mission. The same conditions are applied to the task re-allocation when the target number changes. Figure 7 shows an example of a new purple color target added into the operation. It turns out that C-CBAA assigns the task to the nearest UAVs for the yellow and blue groups. The algorithm selects enough number of UAVs to minimize the mission time with the minimum distance to the new target.

3.  Collision avoidance after reaching the targets: The formation changes to a line formation as agents approaching the target. The followers re-route to the sides of the leader agent so they can increase the surveillance area and execute the mission at the same time. The path planning algorithm in C-CABB will consider collision avoidance during the change of formation. This can be seen from the simulation result of Figure 6b. There is a path crossing between the purple and yellow groups. Our C-CBAA can prevent this possibility of collision and instruct the $P3$ agent in the purple group to change its position to the left when its group approaches to the target. The selected path also avoids the collision within the group by routing from the back of the group. The same path planning strategy can be found from the case of target decreasing scenario as shown in Figure 8b. The $B1$ agent arrives at the center position of the line formation first. It will circle to the left position to avoid the collision with the next arriving agent to the center position.

4.  Reach to target simultaneously and avoid obstacles: The C-CBAA considers the constraints of reaching the target at the same time and avoids obstacles in task allocation during the flight. Figure 8a shows the trajectories of agents before removing

the red target during the movement. The *R*2 agent is closer to the yellow target and easier to run around the obstacle than the *R*1 agent. Therefore, it is assigned to the yellow target after removing the red target. Another example can be seen from Figure 6b. The *R*2 agent changes its position from the back to the left to avoid the possibility of collision to the obstacle. The same can be found in the red group of Figure 7b.

## 6. Conclusions

In this research, we tackled scenarios of multi-targets task allocation with random obstacles in real time movement environment. Two algorithms are proposed under the distributed MAS architecture to achieve the target autonomously. The Committee and Consensus Base Auction Algorithm and Committee-Based Consensus Dynamic Trajectory Planning Algorithm can perform task re-allocation and avoid obstacles while maintaining a compact formation in real time. Simulation results confirmed that the proposed algorithms C-CBAA were able to converge faster than the baseline CBAA methods under various scenarios.

The current task allocation is based on a load balancing method by assuming each agent has the same load during the operation and each agent only assigns one task. In the future, we are considering allowing agents to take more than one task within the time limits. In reality, task allocation should also consider the priority of the task, task requirements and the functional capabilities of each agent. The same for the path planning during the obstacle avoiding process. There should be a weighting associated with the risk for a path option in obstacles avoidance. We plan to allow different weightings to targets as well as obstacles for path planning in the future. We also plan to expand our algorithms to a 3D application such as the operation of UAVs.

**Author Contributions:** Conceptualization, W.-Y.Y., V.-W.S. and Y.-L.L.; methodology, V.-W.S.; software, X.-Q.H. and H.-Y.L.; validation, X.-Q.H. and H.-Y.L.; formal analysis, Y.-L.L.; investigation, W.-Y.Y., X.-Q.H. and H.-Y.L.; resources, W.-Y.Y.; data curation, W.-Y.Y.; writing—original draft preparation, W.-Y.Y.; writing—review and editing, W.-Y.Y. and V.-W.S.; visualization, Y.-L.L.; supervision, V.-W.S. and Y.-L.L.; project administration, W.-Y.Y. and V.-W.S.; funding acquisition, W.-Y.Y. and Y.-L.L. All authors have read and agreed to the published version of the manuscript.

**Funding:** This work was financially supported by National Chung-Shan Institute of Science and Technology, R.O.C (XC08347PE18PE-CS, developing of the algorithm) and Ministry of Science and Technology, R.O.C (MOST 109-2623-E-606-003-D, approbation and scenario application).

**Institutional Review Board Statement:** Not applicable.

**Informed Consent Statement:** Not applicable.

**Data Availability Statement:** Not applicable.

**Conflicts of Interest:** The authors declare no conflict of interest.

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
