# Peer review of "Auction-Based Consensus of Autonomous Vehicles for Multi-Target Dynamic Task Allocation and Path Planning in an Unknown Obstacle Environment"

_applsci, doi:10.3390/app11115057_

Round 1
Reviewer 1 Report
The proposed method describes a method for UAV dynamic path planning and the method is supported by simulation results. Although the paper possesses some novelty and the results are supported, the following issues should be addressed, before the paper can be accepted for publication:
1) Comparative results with other methods are missing. The method described in doi: 10.1109/ACCESS.2013.2283031 shares common goals with the one in this paper, so the authors are asked to provide a comparison study among this and the proposed method.
2) The authors claim that the intended method is for UAV (although suitable for any mobile robotic agent), yet the equations and the simulations refer only to the 2D case. They should justify this in the revised paper.
Author Response
Part I About Review report:
1.Does the introduction provide sufficient background and include all relevant references? (Can be improved)
Response: We have added some relevant references [1, 36, 37] in our introduction and relate work sections. You can check at line 19, 23, 52,164, 203 and 229.
2. Is the research design appropriate? (Can be improved)
Response: We have added some descriptions to explain our research design in the methodology section. You can check at line 229.
3. Are the methods adequately described? (Can be improved)
Response: We have modified the labels and descriptions of Figs. 2-8 to make our research descriptions and experiments clearer. You can check at line 303.
4. Are the methods adequately described? (Can be improved)
Response: We have modified the labels and descriptions of Figs. 2-8 to make our research descriptions and experiments clearer. You can check at line 303.
Part II About the Comments, please see the attachment

Reviewer 2 Report
Undoubtedly, the UAV technology has recently been developed rapidly in a wide variety of applications. I share the opinion that, coordinating a team of autonomous UAVs to complete missions in an unknown and changing environment has been a challenging and complicated task. In this article Authors modify consensus-based auction algorithm so that it can dynamically reallocate tasks among UAVs that can flexibly find a path to reach multiple dynamic targets while avoiding unexpected obstacles and staying close as a group as possible simultaneously.
My comments to the article are as follows:
- As part of the Introduction, I propose to make a wider background in the field of controlling UAV objects, including the fact that attempts are made to use EEG signals to control them. For example, reference may be made to: The Use of Brain-Computer Interface to Control Unmanned Aerial Vehicle, Automation 2019: Progress In Automation, Robotics And Measurement Techniques, Book Series: Advances in Intelligent Systems and Computing, Springer.
- In Fig. 2 the description of the axis is missing. Please introduce additional markings and enlarge the figures. A similar remark applies to Figs. 3, 4, 5 - 8.
- I propose to review the article in terms of punctuation. Sometimes the sign is missing or there are too many of them.
Author Response
Part I Review Report Form:
1. Does the introduction provide sufficient background and include all relevant references? (Can be improved)
Response 1: We have added some relevant references [1, 36, 37] in our introduction and relate work sections. You can check at line 19, 23, 52, 164, 203 and 229.
2. Is the research design appropriate?(Can be improved)
Response 2: We have added some descriptions to explain our research design in the methodology section. You can check at line 229.
3. Is the research design appropriate?(Can be improved)
Response 3: We have modified the labels and descriptions of Figs. 2-8 to make our research descriptions and experiments clearer. You can check at line 303.
Part III Comments and Suggestions for Authors : Please see the attachment.

Round 2
Reviewer 1 Report
None of the points I have raised have been addressed adequately. Nonetheless, I can understand the authors' claims.
Yet, since the method only covers only the 2D case, I suggest that the word UAV should be replaced by robot, through the paper. One way or the other, the authors claim that "the terms "UAV" and "agent" may be used interchangeably without special mention in many situations of the paper", so why not using only one term, i.e. "robot"? Moreover, the authors claim: "the same algorithms can be simply extended into 3D case", well I agree that the algorithm can be extended, but this is not straightforward, thus I insist to remove the acronym UAV throughout the manuscript.
Author Response
Point : Yet, since the method only covers only the 2D case, I suggest that the word UAV should be replaced by robot, through the paper. One way or the other, the authors claim that "the terms "UAV" and "agent" may be used interchangeably without special mention in many situations of the paper", so why not using only one term, i.e. "robot"? Moreover, the authors claim: "the same algorithms can be simply extended into 3D case", well I agree that the algorithm can be extended, but this is not straightforward, thus I insist to remove the acronym UAV throughout the manuscript.
Response: Thank you for your time and suggestions. To extend the concept in a more general way, we agree with your comments and have replaced all “UAVs” wording by either “autonomous vehicles” or “multi-agent” through out the paper to match with the current simulation results in 2D scenarios. We have also added some descriptions to explain the autonomous vehicles to make the paper consistent with the title and descriptions. Please check at lines 18-23, 34-35 and 59-61. All UAVs relate terms had being changed to vehicles or agents corresponding through out the paper.